# Mobilising social support to improve mental health for children and adolescents: A systematic review using principles of realist synthesis

**Annette Bauer**[1]*, **Madeleine Stevens**[1], **Daniel Purtscheller**[2], **Martin Knapp**[1], **Peter Fonagy**[3], **Sara Evans-Lacko**[1], **Jean Paul**[2]

1 Care Policy and Evaluation Centre (CPEC), London School of Economics and Political Science, London, United Kingdom, 2 Department of Psychiatry, Psychotherapy and Psychosomatics, Medical University Innsbruck, Innsbruck, Austria, 3 Division of Psychology and Language Sciences, University College London, London, United Kingdom

* a.bauer@lse.ac.uk

**Data Availability Statement:** All relevant data are in the paper and its Supporting information files.

## Abstract

Social support is a well-recognised protective factor for children's mental health. Whilst many interventions exist that seek to mobilise social support to improve children's mental health, not much is known about how to best do this. We sought to generate knowledge about the ways in which social support can be mobilised to improve children's mental health. We conducted a systematic review, which followed the principles of a realist synthesis. The following databases were searched: PubMed, CINAHL, Ovid MEDLINE, PsychINFO, EMBASE, Child and Adolescent Studies, EconLit and SocINDEX. Studies were included if the age of participants was between 0 and 18 years and they evaluated or described programme theories of interventions that sought to improve children's mental health by mobilising social support. Relevance and quality of studies were assessed, and data were extracted and analysed narratively. Thirty-three articles were included. Studies varied substantially with regard to the detail in which they described the processes of mobilising social support and expected mechanisms to improve children's mental health. Those that provided this detail showed the following: Intervention components included explaining the benefits of social support and relationships to families and modelling friendly relationships to improve social skills. Pathways to improved outcomes reflected bi-directional and dynamic relationships between social support and mental health, and complex and long-term processes of establishing relationship qualities such as trust and reciprocity. Parents' ability to mobilise social support for themselves and on behalf of children was assumed to impact on their children's mental health, and (future) ability to mobilise social support. Although interventions were considered affordable, some required substantial human and financial resources from existing systems. Mobilising social support for vulnerable children can be a complex process that requires careful planning, and theory-informed evaluations can have an important role in increasing knowledge about how to best address social support and loneliness in children.

**Funding:** AB, DP, JP received funding from the Austrian Federal Ministry of Health Science and Research through the Open Innovation in Science Center at the Ludwig Boltzmann Gesellschaft GmbH in Vienna (Austria). The funder was not involved in the study design, collection, analysis or interpretation of data. The others authors (PF, SEL, MS, MK) received no funding for this study.

**Competing interests:** The authors have declared that no competing interests exist.

## Introduction

Social support refers to the extent to which an individual has access to, or perceives they have access to, assistance and resources provided by people in their social network [1]. It is concerned with the function of social relationships rather than their structural constellation, which makes it, to some extent, distinguishable from other concepts such as social capital or social connectedness, although the terms are overlapping and sometimes used interchangeably [2]. Whilst potential adverse effects of social support have been established, too, social support is generally regarded as an important protective factor for positive mental health at all ages, including during childhood and adolescence [2–4]. For children and adolescents, it can be associated with lower rates of depression, generalised anxiety and post-traumatic stress disorders [5–10], suicide [11], behavioural and school adjustment problems and risk behaviours [7,12,13]. Various studies that investigated the association between social support and protection from mental health problems found that sources of support (e.g. informal or formal) vary across the life span [8]. Findings from the youth literature suggest that sources and types or characteristics of social support might influence the magnitude of the protective (or sometimes adverse) effects of social support on mental health, but that important evidence gaps remain [14–16].

In the pursuit of realising potential mental health benefits for children and adolescents, the mobilisation of social support has been incorporated into the design of many interventions [17], either as one of several components, or as the only or main component. Researchers have highlighted the challenges of designing, implementing and evaluating what they call social support interventions due to the multi-dimensionality of the concept, which is defined and measured in many different ways [18,19].

Two main social support theories, the stress-buffering and main-effects models [20–22] have been leading the field for decades. Whilst the stress-buffering model suggests that social support reduces the impact of negative life events on a person's (mental) health, the main effects model hypotheses that there are (mental) health benefits inherent to social relationships irrespective of the stress experienced by a person. Based on those and additional theories, many different pathways and mechanisms have been proposed by which social support is expected to influence mental health [23–25]. They include: creating feelings of belonging, security and self-worth; developing trustful and intimate relationships; adoption of health-related behaviours through social networks; and improving access to resources and opportunities [22,26].

Overall, however, there is not much knowledge on how interventions should be designed to mobilise different types of social support in order to improve children's mental health [27]. This kind of knowledge, including about how different types and sources of social support influence mental health outcomes, which differ according to age group, is important in order to develop programme theories, and understand gaps in evidence [27]. By reviewing the intervention literature, we sought to understand:

1. Ways in which social support can be mobilised in order to improve the mental health of children and adolescents.

2. The mechanisms by which social support is expected to (or has been found to) lead to improved mental health for children and adolescents.

We hypothesised that the following areas would be important to investigate: sources and types of social support; metrics used for measuring social support and mental health; population characteristics. Finally, we wanted to understand resource inputs required for the delivery of interventions, and their potential role in influencing outcomes.

## Methods

We carried out a systematic review of the literature, which followed principles of a realist synthesis [28,29]. Realist review or synthesis is an approach to reviewing evidence on complex social interventions which seeks to provide an explanatory analysis of how and why interventions work (or do not work) in particular contexts or settings and for particular populations. It combines theoretical understanding and empirical evidence, with a focus on explaining the relationship between the context in which an intervention is applied, the mechanisms by which the intervention works and the outcomes produced. Underlying this is an understanding that change is not just generated through the influence of interventions, but through resource inputs, human reaction processes and contextual factors. It is particularly suitable for the development of programme theories [28]. We used principles of realistic review in the inclusion of studies and when extracting data from studies.

We searched for studies concerned with the conceptualisation and evaluation of interventions that sought to mobilise social support to improve mental health of children and adolescents. We were interested in individuals of ages from zero to 18 years. We included infants in the review in order to capture interventions that seek to prevent mental health problems for children by focusing on early childhood.

### Inclusion criteria

We included studies that examined interventions where the mean age was between 0 and 18 years. Studies were only deemed appropriate for inclusion if they described or evaluated interventions that had specific aims to increase social support as indicated by the inclusion of social support into the programme's or study's aims, as well as the inclusion of a measure of social support in the study design. We relied on authors' explicit descriptions of social support. For example, we would not infer from peer support intervention that the intervention was about social support unless the authors discussed social support explicitly. This approach has been used in a global review of active components present in interventions aimed to improve adolescent mental health [30]. Social support could refer to the child's or parent's social support as long as the intervention sought to mobilise social support in order to achieve improved children's mental health, which had to be an explicit goal. No (additional) restrictions were applied regarding type of settings. Primary outcomes were changes in children's mental health. Studies were included if they measured mental health or associated indicators or, for infants, predictors of mental health. This included studies that measured self-esteem, hope or coping for children, and studies that measured mother-infant attachment for infants. We accepted papers that reported on mental health outcomes in previous evaluations (if they were appropriately referred and cited in the paper). Studies also needed to include, as a secondary outcomes, a measure of social support. Outcomes for mental health and wellbeing and social support could use a standardised scale, a sub-domain of a scale, survey or activity data, or be evaluated qualitatively. Since we were interested in various evidence types (including conceptual papers reporting programme theories) we also accepted studies that did not specify outcome assessments but outlined the types of outcomes that could be included in evaluation studies.

Full texts of included studies needed to be in English language. There were no restrictions in terms of their study design; we included experimental, non-experimental, qualitative, and mixed-method designs, evaluation protocols and conceptual papers reporting programme theories.

### Exclusion criteria

We excluded studies of interventions that were seeking to improve parental behavioural outcomes but did not mention children's mental health in their programme goals. Consequently,

we excluded studies of interventions that were only concerned with reducing child maltreatment. We excluded populations exposed to traumatic events or extreme adversities such as war, natural disasters, epidemics, and terrorist attacks. We also excluded studies that specifically targeted children with autism or severe communication needs.

## Search strategy

Search terms that described the population, social support, and intervention were initially scoped on PubMed before a revised search strategy was developed for PubMed. The search strategy was adapted for each of the following databases: CINAHL, Ovid MEDLINE, PsychINFO, EMBASE, Child and Adolescent Studies, EconLit and SocINDEX. Searches identified studies between 01/01/2008 to 08/06/2018. An example of our search strategy is provided in the electronic material (S1 Box).

## Study selection

Fig 1 shows the PRISMA flow chart of the screening process. Titles and abstracts were assessed by one reviewer (AB). Articles that clearly did not meet criteria were rejected at this stage. Full texts were retrieved for potentially relevant articles. The same reviewer (AB) screened studies based on full text. Studies where it was unclear whether inclusion or exclusion criteria were met were subject to a detailed screening process undertaken by four reviewers (AB, DP, JP, MS); this involved completing a screening tool, and various rounds of discussions.

## Assessment of relevance and quality

Following guidance for realist reviews [29], studies were appraised as to their relevance as well as their rigour. The relevance of the study was assessed based on the extent to which the study defined, conceptualised and measured social support, and explained how it was mobilised and expected to improve children's mental health outcomes. Using the latest version of the Mixed Methods Appraisal Tool [31] the study rigour was assessed in relation to choice of study design, sample size, data collection methods, and outcomes. Following the guidance and algorithm provided by the tool, we applied 'low', 'high' and, where information was insufficient to rate the criterion, 'can't tell' ratings. The algorithm provides quality criteria as well as examples of how to apply those for qualitative, quantitative randomised controlled trails, quantitative non-randomised controlled trials, quantitative descriptive studies and mixed-method studies. Studies were not excluded based on relevance or rigour. Instead, the rating informed the interpretation of findings.

## Data extraction, analysis and synthesis

Data were extracted from all sections of papers using bespoke forms and analysed narratively using headings of a realist synthesis and categorised into age groups of children. Age categories included infants aged 0 to 2 years, children aged 3 to 9 years, and adolescents aged 10 to 18 years. For studies, where the age range fell between two categories, they landed in the category that captured more years; e.g. if the inclusion was 5 to 12 years, the study would land in the 3 to 9 years category. By identifying data patterns, a realist synthesis seeks to derive information about relationships between resource inputs, human reaction processes, and contextual factors for interventions or intervention components, and how those lead to particular outcomes. In this paper, our main interest was to understand how social support was conceptualised, e.g. with regard to types of social support, which changes in human interaction processes were assumed to be required in order to improve children's mental health outcomes, and how those

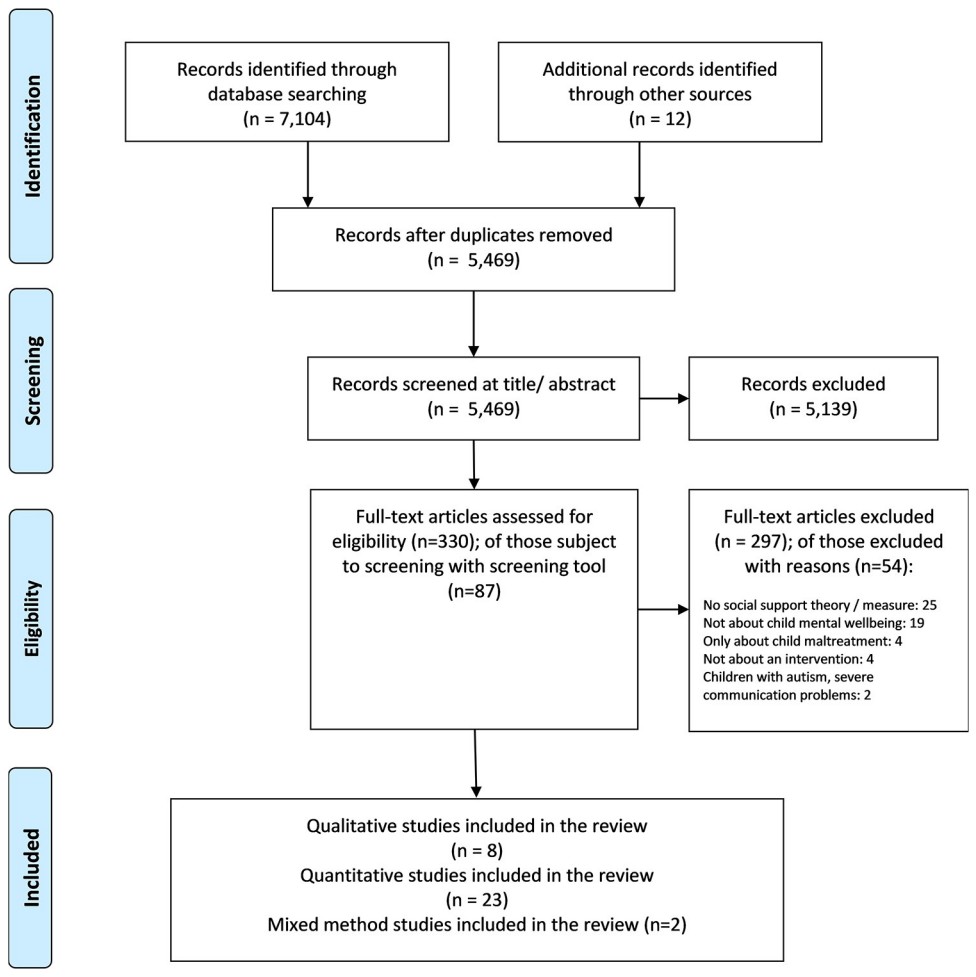

**Fig 1. Flow of studies into the review.**

were modelled into the intervention design. We used the above-mentioned dimensions (intervention components, context, mechanisms, and outcomes) for the synthesis of the data and we present findings by research questions.

## Results

Thirty-three studies were included [32–64]. S1–S5 Tables in the supporting information provide details of the studies including the details of how assessments of study relevance and quality were derived. In many studies, social support was not well-conceptualised, and many studies were weak in explaining how social support was mobilised or expected to lead to improved children's mental health. Most studies did not specify the types or sources of social support they sought to address or the rationale for doing so. Social support sometimes only referred to a single source of support such as health professionals, peers or mentors, parents or school staff. Interventions most commonly mobilised the social support of parents, followed by those studies that were about mobilising social support of children. Only a few were about increasing social support of the family as a whole.

More than a third of the interventions were mentoring, peer support, or a combination of the two. Other interventions included parenting education, training or support (covered in

seven studies), complex family support involving goal-setting and support-planning, linkage projects with schools and other public services, community capacity-building and service integration approaches, and psychoeducation or mental health literacy training. More than half of the programmes were delivered in the form of individual (family) support, and the rest were delivered in the community in group sessions or in mixed formats. Two studies were delivered via telephone or internet. Most interventions were provided by volunteers, community workers or psychological therapists. Only four studies [43,49,51,56] were conducted in middle-income countries (South Africa, India and Pakistan), whilst the rest were from high-income countries (North America, Europe, Australia, Japan). Tables 1–3 provide details about studies in relation to their programme components, contextual factors, population, mechanisms for improving children's mental health, and findings on outcomes.

### Age-group specific findings: Infants (0 to 2 years)

**How is social support mobilised, to which populations, and in which context?** Four studies [33–36] were concerned with providing or mobilising social support for parents of infants to improve children's mental health (S3 Table). Two of the interventions were linking parents with their community through a trusted lay person, who would connect the parent with informal and formal support [34,35]. One intervention was linking mothers with health professionals [36]. Two of the interventions included teaching skills such as mother-infant interaction [33] or broader social skills that would allow mothers to build relationships [35]. In the study by Mitchell et al [35], a mentoring mother modelled friendly relationships and helped to create opportunities for the mothers to practice newly gained skills together in the community. All four interventions addressed some form of informational support in regards to parenting; one intervention addressed informational support only [36], whilst one specifically addressed all types of social support (i.e. emotional, practical and informational support) [34]. The study by Stubbs and Achat (2016) [34] was the only one that targeted the whole families, whilst the others targeted mothers. All four took place in high-income countries. They targeted parents at risk of social isolation, stress and mental health problems (Table 1). One intervention targeted women with postpartum depression [33]. In three studies [33,35,36], mothers received the intervention alongside professional (mental) health services (Table 1).

**Does an increase in social support lead to improved children's mental health, and what are the mechanisms by which this is (expected to be) achieved?** Three studies [34–36] reported increased social support, which in two studies referred to social support from health professionals, the community or formal services not measured with standardised scales (Table 1). None of the studies reported an increase in support from partners, parents or friends (even though all four studies measured this). One moderate-quality study [33] that captured perceived social support using a standardised scale reported findings in relation to social support that favoured the control group. Authors explained this as follows: mothers in the control group formed their own networks that were more sustainable than the relationships formed by mothers in the intervention group with volunteering peers. In additionthe matching of volunteers to mothers was considered inadequate and the teaching component of the intervention might have hindered the development of equal and trusting relationships between peers and mothers.

Studies described how social support was expected to protect against negative impacts of depressive symptoms and stress, improve mother-infant interactions, parental self-efficacy, agency, and self-esteem (Table 1). In turn, those would allow parents to form new relationships, and this would improve child cognitive and social development, children's ability to form social relationships in the future as well as improve child behaviour and wellbeing. Infant

**Table 1. Information about programme theories and findings from included studies concerned with infants (0–2 years).**

| Study ID | Intervention components | Context | Mechanisms for improved child mental health | Social support outcomes | Child outcomes |
|---|---|---|---|---|---|
| Cho et al (2013) [36] | Building supportive relationships with healthcare providers and offering informational support | Mothers with pre-term infants<br>High levels of stress, low levels of personal networks<br>Provided alongside health professionals | More secure mother-infant relationship (and reduced stress), which improves child development outcomes including child's ability to form social relationships | Mothers in intervention group reported significantly more social support from healthcare professionals but not from partner, other parents or friends | No positive effect of intervention on child development |
| Letourneau et al (2011) [33] | Teaching maternal-infant interaction and providing social support | Women with postpartum depression<br>Provided alongside professional treatment and support | Social support protective against depressive symptoms and stress, enhances maternal–infant interactions, and subsequently, infant's cognitive and social development | Findings with regards to social support favoured the control group | No significant effect of intervention on infant cognitive and social development |
| Mitchell et al (2015) [35] | Teaching about relationships through modelling friendliness, openness, trust, honesty, respect (authenticity of perceived mutuality)<br>Linking mothers to community; providing opportunities to practice being with other people, helping them to engage with other people | First-time mothers (described as more likely to accept help)<br>Patterns of intergenerational child abuse and extreme social disadvantage<br>Provided alongside professional support | Strengthened self-esteem, self-confidence and agency, which helps mother to benefit from formal support and build new forms of informal support (thus bridging between informal support and formal support), which in turn improves child development | Increased access to and interaction with the formal health system and local community | Reduced risk to infants and increased attachment of mothers to their infants |
| Stubbs and Achat (2016) [34] | Linking parents to community networks<br>Offering formal support and offering emotional, practical and informational support<br>Treating parents as partners in their children's care and building on their strengths | Disadvantaged areas (suburbs)<br>Families experienced risk factors categorised as vulnerable or complex | Improved parental self-efficacy leading to improved child behaviour and wellbeing | Parents more likely to provide and receive some form of social support at follow up (e.g. church or other community groups) but not all relationships significant (informal more successful than formal) | No improvements in infant health or development (but improved parental self-efficacy) |

outcomes measured in studies included infant-attachment, and socio-emotional and cognitive development (S3 Table). Only one low-quality study [35] reported positive effects on mother-infant attachment (and improved parenting skills). Two studies [34,36] reported that design errors might have explained the lack of evidence on infant outcomes.

## Age-group specific findings: Children (3 to 9 years)

**How is social support mobilised, to which populations and in which context?** We identified thirteen studies [32,37–48] in this category (S4 Table). This included two studies [37,39] that did not specify the age, but where we inferred from the background information that they referred primarily to children in early or mid-development ages. Most interventions aimed to increase parents' social support by: directly providing social support, increasing access to services, reinforcing to parents the importance of social relationships and teaching relationship or help-seeking skills (Table 2). For a few interventions, this specifically referred to improving relationships with childcare institutions or schools. Three interventions sought to change capacities of social networks and whole service systems (including schools) to mobilise social support for parents [40,47,54]. Some interventions focused on increasing positive emotions such as hope and self-esteem, which were expected to lead to development of new relationships. Studies focusing on changing the perception of parents about social support

**Table 2. Information about programme theories and findings for included studies concerned with children (3 to 9 years).**

| Study ID | Intervention components | Context | Mechanisms for improved child mental health | Social support outcomes | Child outcomes |
|---|---|---|---|---|---|
| Ayton and Joss (2016) [32] | Teaching parents social and parenting skills to develop relationships with others and establishing community connections and social networks<br>Addressing social determinants of health and remove barriers<br>Offering practical and emotional support | Vulnerable and isolated parents affected by intergenerational poverty (excluding those with unmanaged violence, debt, etc.) | Improved parenting skills leading to child health and wellbeing | Parents experienced improvements in social support provided by the mentor (in addition to other improvements in employment, housing, mental health, drug and alcohol use) | Improved parent-child relationships (because of increase in parental emotions and skills)<br>Improvements in social determinants of child (mental) health (but no evidence on child mental health presented) |
| Branch et al (2013) [38] | Broadening understanding by professionals of interconnectedness and interdependencies of child's life<br>Improving strength of relations between different levels of organisations (mainly schools) involved in child's life | Most families alienated from school; language barriers, cultural factors; historically disastrous experiences with government for indigenous people<br>Children with medium to high needs; withdrawn or aggressive behaviour; health problems; isolation related to language | Mutual responsibility among professionals to improve child wellbeing leads to innovative solutions that are expected to improve child wellbeing | New relationships especially between schools and families as parents lost fear of institutions and started to build trust (collected via prompts about connections in qualitative interviews) | Evidence of adjustments made by school, parents and programme staff with benefit for child's behaviour and school attendance (but no evidence on child outcomes presented) |
| Byrne et al (2012) [37] | Reinforcing parents' perception of social support and increasing their satisfaction with social support networks<br>Teaching parenting skills | At risk families often without mutual supportive relationships including with partner<br>Parents have negative perception and distrust towards services and are less likely to accept formal support | Improved parenting skills assumed to improve child development and wellbeing<br>Parent satisfaction with formal and informal support hypothesised to increase help seeking behaviour including help for child development support | Parents' increased satisfaction with and use of informal and—to a lesser extent—formal support (e.g. neighbourhood associations, child welfare support) | Improved parental outcomes (e.g. agency) linked to increase in perceived social support (but no evidence on child mental health presented)<br>Some evidence that stronger positive effects of informal support and of negative effects of 'too much' formal support on parental agency |
| Doty et al (2017) [39] | Teaching parenting skills<br>Building positive emotions<br>Increasing confidence of parents to mobilise needed support for child and build social capital for benefit of child health and academic achievements | Economic disadvantaged families with certain level of extant social capital | Children's develop early socioemotional skills due to increased social capital of parents, which help them to build or have access to supportive social networks, which in turn is associated with psychological and social adjustment in later adulthood | Improved relationships among family members and between family members and social networks | Improved school attendance and grade promotion; fewer risk behaviours |
| Drummond et al (2014) [40] | Service integration to increase access to formal support for families<br>Parental involvement with childcare and schools hypothesised to lead to richer social support networks | Low income families, including aboriginal and other minorities populations; families on government assistance programme | Family functioning hypothesised to influence family linkage to services and health outcomes<br>Better school achievements for children when parents more involved with schools<br>Child engagement in recreation activities hypothesised to lead to improved child wellbeing | N/A (protocol) | N/A (protocol) |

*(Continued)*

**Table 2.** (Continued)

| Study ID | Intervention components | Context | Mechanisms for improved child mental health | Social support outcomes | Child outcomes |
|---|---|---|---|---|---|
| Eddy et al (2017) [41] | Provide opportunities for child to participate in enriching experiences that enhance ability to envision a positive future Providing social support opportunities for children—like access to academic assistance and health care | Children at risk; living in disadvantaged areas | Children build and engage in social relationships with others including peers, teachers, parents; this is expected to improve child social-emotional, cognitive and identity development | Increased received social support from mentors (in form of long-term relationships) | Significant effects in terms positive child behaviour and less trouble in school; and trend for higher child behavioural and emotional strengths |
| Ingram et al (2015) [42] | Teaching parents skills how to asks for and utilise social support (e.g. active listening, modelling, guided practice) | Socially isolated families, many have child protection record (children still living with parent); exposed to multiple stressors | Increased parental capacity and improved family interactions expected to reduce child behaviour problems and improve school attendance and achievements | Increased social support available to parents and improved family relationships | Improved child well-being (moderate effects), in addition to improved school attendance and reduced youth crime |
| Lachman et al (2017) [43] | Teaching parenting skills and non-violent behaviour towards children | Low and middle income country context with high rates of HIV, drug and alcohol addictions and violence Intervention provided by low skilled staff | Increased parent's social support and self-efficacy expected to reduce risk of child maltreatment and to lead to improved child behaviour and socio-emotional regulation skills (role of social support not well described) | No significant differences in parent's perceived social support | Negative effect on child behaviour |
| Marcynyszyn et al (2011) [44] | Teaching parenting skills (in particular managing child behaviour) | Parents involved in child welfare system | Satisfaction with support provided is assumed to influence child outcomes through retention in programme (role of social support not well described) | Parents reported higher levels of family support, and (to a lesser extent) friend support Perceived helpfulness of resources largely unchanged other than for parenting group itself | Improved child behaviour; reduced child difficulties (small effects) |
| Nabuco et al. (2014) [45] | Teaching parenting skills to seek for support for child Providing opportunities for parents in the same community to discuss information and ideas, share experiences, offer support | Families in poverty and lack of social networks and support; children lack bonding with parents; low self-esteem; poor literacy/ numeracy skills; majority of children did not attend any preschool programme | Increased parenting knowledge, empowerment and resources for educating children leads to better cognitive and social development | Higher social support perceived by parents in the intervention group | Improvements in child cognitive and social development |
| Pancer et al. (2013) [46] | Providing information about community services and resources Changing the environment | Areas with substantial neighbourhood disadvantage and significant risks for child development | Skills to access community resources Parents feeling sense of connection with others in their community Child benefits from increased access to support | No significant changes in: parent-reported social support; parent involvement in social activities; youth community involvement | Not differences in child behaviour problems or social skills |
| Parcel and Pennell (2012) [47] | Joint planning to support children in school Participatory decision making and trust building processes with parents Strengthening linkages within the family and linkages to school and community organisations | Low-income families and neighbourhoods, black and ethnic minorities, children at risk of academic failure | Increase in family and school social support hypothesised to predict academic achievement and social adjustment and behaviour outcomes for children | Improved relationships within families | Improved mental health, academic achievements, reduction in youth crimes; improved family functioning |

(*Continued*)

**Table 2.** (Continued)

| Study ID | Intervention components | Context | Mechanisms for improved child mental health | Social support outcomes | Child outcomes |
|---|---|---|---|---|---|
| Vazquez et al. (2017) [48] | Teaching parents about child development | High proportion of immigrants, mainly from Latin America; described as having complex lives; parents report feeling isolated | No hypotheses stated | Increased social support as perceived by parents; parents no longer feeling isolated in their parenting role. Parents viewed the program as source of social support because of program contents, facilitator strategies, support from parents, and by institutions and community | Reduced negative child behaviour and increased school performance |

described how interventions were increasing parents' satisfaction and trust with public institutions by providing a trusted person, who would facilitate those links (see for example Drummond *et al.* [40]). Some studies assumed that children of parents with increased social support would acquire new socio-emotional skills, thus allowing them to build their own social support systems in the future, highlighting the intergenerational effects of social support (see for example Doty *et al.* [39]).

All but one study [54] targeted children and their families experiencing socio-economic disadvantage, including children of parents with mental illness from migrant, black or ethnic minority backgrounds. The study by Hauken et al [54] targeted children whose parents were living with cancer. One study [43] took place in a low-income country with high rates of HIV, substance abuse, and violence, whilst all other studies took place in high-income countries. Studies described families' social isolation and lack of social support, which could include their alienation from school and public services, due to distrust towards government, based on their own past, or intergenerational experiences as a community (Table 2). Studies described problems experienced by children, which included behavioural and health problems, poor literacy and numeracy skills, low self-esteem, lack of bonding with parents, and academic underachievement. Two studies referred specifically to families involved with the child welfare system.

**Does an increase in social support lead to improved children's mental health, and what are the mechanisms by which this is (expected to be) achieved?**   The vast majority of studies reported increases in parents' social support, which referred most commonly to improved family relationships, and to a lesser extent, to other parents, and improved relationships between families and schools (Table 2). Only one study [41] referred to social support as mobilised by children directly, whilst all other studies referred to social support as mobilised by parents (and teachers) on behalf of the child. In some studies, social support was reported as an outcome of the implementation of the intervention, referring for example to mentoring or peer support, whilst in other studies it was reported as a primary or secondary outcome.

The majority of studies reported improved child behaviour, cognitive and social development outcomes, alongside improved school performance or attendance, as well as improved coping, psychological functioning or help-seeking (Table 2). Some studies reported that effects were only small, and two studies [43,62], including a high-quality one, reported negative effects on child behaviour, emotional problems or school adjustment. One study explained this as short-term negative emotions when opening up about painful experiences, whilst the

**Table 3. Information about programme theories and findings of studies concerned with adolescents (10–18 years).**

| Study ID | Intervention components | Context | Mechanisms for improved child mental health | Social support outcomes | Child outcomes |
|---|---|---|---|---|---|
| Asghar et al 2018 [49] | Building life skills of girls and training their caregivers and service providers in supporting girls | Displaced and host-community adolescent girls; some living in camps or with restricted movement in public; exposed to gender-based discrimination and stigma | Social support networks together with self-esteem and hope (human assets) and physical assets hypothesised to protect from future violence Relationship to a mentor (trusted adult) expected to lead to greater resilience | Higher odds of trusted non-familial female adult and friend; girls report increase in trust to friends No change in having a person in community to talk to in case of sexual violence; no change in quality of relationship with caregiver | Increase in self-esteem and hope |
| Bohleber et al (2016) [50] | Building positive peer support culture Providing information about mental health promotion Facilitating exchange about such information and access to support | Young people who enter work life early (due to national system requirements) and unemployed young people; both described as major stressors | Increase perceived social support hypothesised to lead to stress reductions | No effects found on perceived social support | Reductions in child behaviour problems; no effects on stress |
| Cluver et al (2017) [51] | Teaching social learning and parenting skills involving both parents and youth Teaching about identifying external support Linking parent with another parent in program | Low literacy populations in rural settings; one of the poorest provinces; implementation in local language; no participant exclusion criteria | Increased social support expected mediator for positive parenting behaviours | Large effects for increased access to social support for parents and adolescents | Significant reduction in child behaviour problems 'rule breaking behaviour' and 'aggressive behaviours' |
| Deutsch et al (2017) [52] | Supporting youth identity and development through modelling relationship building and (social) skills and providing safe place for opening up and practicing skills Providing informational support (guidance and advice) | Girls at risk (emotional, academic, social) and not receiving other (formal) support; majority receive free or reduced lunch (lower socio-economic status) | Improvement in social skills, trust in relationships expected to improve child psychosocial outcomes (including academic outcomes) | Increased interacting and deepening of relationships with peers Increased skills in developing new social relationships Increased social relationships outside the group | Improvements in academic and self-esteem domains (but not in social/relational domains) |
| DeWit et al (2016) [53] | Modelling effective adult communication and pro-social behaviour (including praise), teaching skills and offering intellectual challenges through educational and recreational activities Providing experiences of close and secure attachment | High proportion living with single parent; substantial proportion not living with their biological parents; and from ethnic minority groups | Greater perceived value of interpersonal relationships, increased social skills, emotional regulation, coping and confidence expected to allow youth to seek for and engage more effectively in relationships | Increase in perceived emotional support from parents and peers Short-term relationships and re-matching had negative effects on perceived quality of relationships (stronger effects for boys than girls) | Reduction in anxiety and depression, behavioural problems (for those who stay in mentoring relationship for a year or longer) |
| Hauken et al (2015) [54] | Psychoeducation of social network members: Increasing their understanding of the situation, coping strategies and promoting open communication | Families living with parental cancer and dependent children; children often exposed to decreased parental capacity and highly involved in domestic tasks | Direct and indirect effects of family social support network on children's quality of life and mental health expected (indirect ones are via increase in parental capacity, parents' quality of life and mental health) | Not applicable | Not applicable |

(*Continued*)

**Table 3.** (*Continued*)

| Study ID | Intervention components | Context | Mechanisms for improved child mental health | Social support outcomes | Child outcomes |
|---|---|---|---|---|---|
| January et al (2016) [55] | Promoting positive attitudes towards building social support networks and seeking professional help Providing social support Removing access barriers to social support | Parents of children with emerging behavioural and emotional difficulties Families are vulnerable i.e. single parent households, ethnic minority, low socio-economic status | Increased perceived benefit by parents from engaging with services and increased ability to navigate community and school system expected to benefit child's mental health Increased parental efficacy, reduced stress, help seeking behaviour expected to improve child behaviour | Significant increase in perceived social support and concrete support Parents started talking about importance of social support and engaging with services but not about partnering with teachers and schools and not about supporting success of children at home | No improvements in stress, anxiety or depression |
| Leventhal et al (2015) [56] | Resilience-building through facilitated sharing of experiences; goal setting and planning; practicing skills (problem solving, communication) | Low- and middle income country context; high poverty, rural school setting; girls at particular risk of gender-based discrimination | Strengthened psychosocial assets (e.g. coping skills, self-efficacy, social skills, beliefs about helping others) expected to increase social wellbeing (= connections with peers) and psychological wellbeing | Significant increase in social support and peer support | Increased emotional resilience, self-efficacy and psychosocial and social wellbeing (but no effects on depression; and small non-significant effect on anxiety) |
| Romjinders et al. (2017) [57] | Increasing social support Providing accepting and tolerating environment | Youth from sexual and gender minority groups, which are described as more likely to have low levels of perceived social support because of intolerance they experience | Social support as buffer for a non-supportive environment is expected to increase health and well-being Changes in perceptions of social support, increased sense of belonging, perceived control, self-efficacy and self-esteem expected to reduce stress and improved (mental) health | Youth seeing group as family where they can develop trust and be themselves; and have new social relationships with others Some improvements in social support from family | Increase in self-esteem and confidence (evidenced for example in ability to leave an abusive relationship) |
| Schwartz et al. (2013) [58] | Providing training and structure for relationships between youth and a caring adult Providing various types of social support Addressing shortage of naturally forming mentoring relationships, and limitations of regular mentoring (e.g. not same social context; limited availability) | Youth who have dropped out or been expelled from high school | Increased skills of youth to utilise and seek for social support and long-term and stable relationship with caring adult expected to lead to positive youth development outcomes including mental health | Improved relationships of youth with others All social support types provided by mentors over long-term period of time | Positive youth development outcomes e.g. improved self-concept; no change in some risky behaviours |
| Swenson et al. (2010) [59] | Teaching parenting skills and non-violent behaviour Comprehensive assessment of needs; goal setting and planning for wide range of supports to meet complex needs of family | Physically abused youth and their families; large majority are Black and involved with child protection services | Social support in social ecological model hypothesised to reduce risk of child abuse through increased parenting skills and changes in behaviour | Improved informal social support of parents, which lasts beyond intervention | Reduction in mental health problems |
| Van Dam et al. (2017) [60] | Stimulating shared decision making between families, their social network and professionals Providing various types of social support | Adolescents with complex needs at risk of out-of-home placements; difficulties to establish positive natural relationships due to low self-esteem, lack of trust and social skills deficits | Social support expected to increase resilience and reduce stress (social support as buffer against stress) as well as to stimulate to care for oneself (but not further specified) | Majority of youth able to identify a natural mentor from their social network; primarily social emotional support provided by mentors | Reduction in psycho-social problems |

(*Continued*)

**Table 3.** (Continued)

| Study ID | Intervention components | Context | Mechanisms for improved child mental health | Social support outcomes | Child outcomes |
|---|---|---|---|---|---|
| Van Voorhees (2008) [61] | Teaching youth about mental health problems, coping strategies and activation of social support networks and relationships skills | Young people with persistent sub threshold depression but without diagnosed mental health condition | Increased social support through improved perception and acceptance of peer support expected to have buffering effect against developing serious depression for those at risk | Increase in perceived peer support but no changes in perceived family social support (other than 'closeness to mother') | Reduction in depressed mood |
| Valdez et al. (2011) [62] | Teaching families about impact of parental mental illness and coping strategies Teaching families about building external social supports | Children whose mother has depression; mothers recruited from mental health outpatient clinics and judged by clinician as well enough to focus on their families; majority of mothers unemployed and single | By providing emotional and instrumental support to mothers it expected that children's mental health improves | Mothers reported small to moderate improvements in perceived social support | Small decrease in internalising and behaviour problems Moderate improvements in coping and support seeking strategies Increase in emotional and behavioural problems |
| Valdez et al. (2013) [63] | Teaching families about impact of parental mental illness and coping strategies | Mothers are Latina immigrant with depression; low socio-economic status; social isolation; multiple stressors Children experience high rates of suicide attempts, drug use and delinquency, school dropout, early sexual involvement | Parents' improved social support trough healthier marital relationships and parenting competence and skills expected to increase child coping skills and efficacy | Increased mothers and caregivers' perceived social support mainly because of increased marital and family support | Decreased conduct and hyperactivity problems Improved psychological functioning, coping |
| Vella et al. (2018) [64] | Promoting protective factors for mental health by raising awareness, providing information, offering support and educating parents and sport coaches about supporting male youth | Male youth engaged in sporting clubs | Increased help-seeking for professional help and information for mental health problems expected to improve youth mental health | Not applicable | Not applicable |

other explained this as insufficiently skilled staff, who did not have child development knowledge. Two studies [32,37] did not report child outcomes but reported improvements in parental agency or parent-child relationships.

Mechanisms by which social support was expected to improve children's mental health referred primarily to an increase in parents' social support (Table 2). A few interventions were specifically designed to teach parents to ask for and utilise social support, which in turn was expected to improve parental capacity, improve family interaction and reduce child behaviours problems. In some studies, increased access to informational support, better links to schools and other services were considered to lead to improved child development and wellbeing. One study [39] explained this link with children's ability to develop socio-emotional skills that would support their psychological adjustment and ensure access to social support networks in the future. Social support was seen as providing opportunities for experiences that would allow children to build and engage in social relationships, for example by engaging in recreational activities.

## Age-group specific findings: Adolescents (10 to 18 years)

**How is social support mobilised, to which populations and in which context?** Sixteen studies [49–64] focused on providing or mobilising social support to improve adolescents' mental health (S5 Table). Interventions mobilised social support by: modelling healthy

relationships and social skills; offering safe spaces or opportunities for young people to practice their social skills; encouraging youth to seek help for social support; or changing perceptions of the benefits of social support. Interventions sought to provide various types of social support including informational (e.g. how to seek a job), material (e.g. borrowing a car), or emotional (e.g. how to leave an unhealthy relationship). A few interventions referred to providing social support to parents or whole families, or supporting them in developing social support networks. This included providing information about social support, reinforcing the importance of social support, or removing access barriers to social support (Table 3). Other intervention characteristics were primarily educational, e.g. in the form of psychoeducation or self-management. Most interventions applied empowerment and strengths-based approaches towards education. Twelve studies referred to youth exposed to a number of risk factors such as living in poverty, in single-parent households and being treated unequally because of ethnicity, sexual orientation, or gender (Table 3). Youth had low literacy skills, dropped out of or had been excluded from school, had been or were at risk of being removed from their families, or experienced mental health problems, discrimination or abuse. Two studies were about universal preventative interventions, which addressed transition to employment and mental health of young male athletes. Three studies [49,51,56] took place in low- and middle-income countries.

**Does an increase in social support lead to improved children's mental health, and what are the mechanisms by which this is (expected to be) achieved?** More than half the studies reported evidence of an increase in social support, referring mainly to perceived social support measured with standardised scales. This commonly referred to specific types or sources of support such as by families, peers or mentors. Two studies reported no effects on perceived social support [46,50] and one study [53] reported negative effects among boys when the intervention (mentoring) resulted in relationship break-ups between mentor and mentee.

Most studies that reported positive effects on social support also reported positive effects on depression, anxiety or behaviour, or on indicators of mental health such as self-esteem, self-efficacy, coping, hope or resilience (Table 3). Often, those changes were reported alongside improved school attendance, performance or functioning. However, a number of studies reported mixed findings (that is, some mental health outcomes improved, but others did not) or small effects. One study [43] found that child behaviour problems could become worse, which they attributed to insufficiently skilled staff. Eleven of the sixteen studies were of either moderate or high quality.

A range of expected mechanisms for adolescent mental health referred to protective or buffering effects of social support, whilst others referred to social-cognitive effects of social support, such as sense of belonging, identity, self-esteem, self-control and self-regulation, or to relationship aspects such as trust or sense of connection (Table 3). One study [57] described in detail the types of social support provided by different sources of support and hypothesised that peers were more appropriate for providing emotional support, and mentors more appropriate for providing advice and guidance (which was also supported by their findings). Mechanisms for interventions that targeted parents' (rather than youth's) social support included changes in parenting attitudes, behaviours, knowledge and skills, as well as an increased perceived benefit of social support and ability to navigate services for the young person One study [60] hypothesised that the intervention stimulated positive effects of social support on mental health, because of an increased social stimulation to care for one self.

**Resource inputs to deliver interventions.** Studies varied substantially in the detail reported on resource inputs, costs of programmes, or resource implications. Overall, there was not enough information to carry out systematic data extraction and analysis. However, we identified some relevant information and common themes. A third of the studies explicitly highlighted the affordability and scalability of interventions, with several studies even

expecting a positive return-on-investment, for example because of expected reductions in criminal justice costs linked to improvements in delinquency and criminal behaviour. Whilst most interventions were provided by volunteers or low-skilled staff, and only included a couple of days of training, thus suggesting low cost of programme delivery, there was also evidence that interventions could require high levels of (unplanned) resources, including substantial time inputs from staff employed by public sector agencies [38,42,47]. Implementation challenges were potentially driving up costs, in particular in areas in which prevention had low priority and staff were sceptical of the value of the intervention [44,45]. One study reported that, in order to remove access barriers for parents and families, additional investment was required to fund travelling costs and childcare [55]. A few studies highlighted that the interventions should not replace existing support for vulnerable populations, but be provided alongside professional support [35]. Only one study reported intervention costs, and those were USD 10,000 to 12,000 per child per year [41]. Some papers discussed whether interventions could be effectively provided at low cost [33,43].

## Discussion

This review synthesised knowledge about how social support can be mobilised through interventions that seek to improve children's mental health outcomes. It is hoped that this knowledge will be useful for practitioners or researchers who seek to develop, implement or evaluate interventions in this area.

### Discussion of main findings across age groups

Our review found that social support was not well-conceptualised in intervention studies, and studies were generally weak in explaining how social support was mobilised and expected to lead to improved mental health for children. Most studies did not specify the types or sources of social support they sought to address or the rationale for doing so. These limitations have been identified previously [19,20]. Studies that did have a more detailed programme theory in relation to social support were describing the process of mobilising social support as complex, dynamic and long-term. They described various components of this process, such as educating children or parents about the benefits of social support, offering repeated opportunities for practising social skills and for experiencing the benefits of positive relationships through reciprocity and trust-building. The importance of such processes has been confirmed in studies which found that relationship satisfaction and reciprocity of relationships are important contributors to improved mental health [65] and reduced loneliness [66]. Some of the identified studies theorised a complex interaction between social support and mental health, in which social support could be a means to positive mental health, as well as the outcome of processes in which aspects of mental health (e.g. self-esteem) were improved, and this led to a capacity to engage further in social relationships. In the field of social neuroscience, underlying cognitive or biochemical processes have been found that seek to explain this bidirectional relationship [67–69]. They suggest that certain mental capacities or cognitive abilities are required in order for a person to see the value of, and engage in, social relationships and in collective actions (so-called 'we intentions') [68]. Those are likely to be diminished for people experiencing prolonged lack of social support and loneliness due to changes in the nervous systems and in gene expression [70], which can trigger fear-based responses to situations, thus leading to erosion of trust in relationships and further isolation [67,71]. Our review also found that most interventions specifically targeted families from low socio-economic backgrounds, but studies did refer to potentially different mechanisms between social support and mental health for this population as identified in the literature [22].

## Discussion of findings by age groups

**Infants (0 to 2 years).**  Interventions in this category tended to be provided alongside (mental) health services to mothers at risk of stress or mental health problems during the perinatal period. They sought to address information needs alongside other support needs, and to help mothers engage with and improve their relationships with (health) professionals. Authors of these studies expected that increased social support for mothers (and fathers) would improve child social and emotional competence either through social-cognitive (e.g. parental self-efficacy) or stress-buffering mechanisms. Small et al (2011) [72] found a lack of impact of social support interventions in this area, which they explained with their focus on information support (i.e. parenting education) rather than companionship, emotional and appraisal support. Similarly, Milgrom et al (2019) [73] highlight the importance of providing different types of social support at different time-points during pregnancy and after birth. As suggested by a high-quality study in our review [33], professional-like advice was potentially crowding-out feelings of trust and self-worth, and naturally evolving relationships, suggesting therefore the challenge of mobilising social support sustainably. Evaluation challenges prevented us from deriving conclusions about whether social support provided to parents during the perinatal period improved children's mental health.

**Children (3 to 9 years).**  Interventions in this category described the social isolation of families, who had very limited formal support from public institutions such as childcare facilities or schools. Several interventions focused on rebuilding such relationships and transforming them from one based on power imbalance to one that was reciprocal and built on trust. Authors of studies expected that by improving those relationships, parents would start engaging in and enjoying child-centred activities, thereby leveraging social capital for the benefit of their children, which in turn would improve children's long-term wellbeing. Another set of interventions focused instead on social support as a protective factor for improved parenting practices and capacities, which in turn was expected to improve family functioning and contribute to improved child development.

In this review two intervention types had the potential to achieve positive child behaviour. One focused on changing bi- or multi-directional relationships involving families and professionals (and sometimes wider communities). The other focused on parents' behaviour., It has been argued that only the first follows a truly ecological model of shared child responsibility supported by international legislation of child rights [74].

**Adolescents (10 to 18 years).**  Interventions included in this category sought to reduce major risks for vulnerable groups, in particular with regard to school failure and risky life choices. Vulnerabilities of youth related to sexual orientation, mental health, and their exposure to discrimination, violence and abuse. Social support was mobilised by providing opportunities for learning and practising social skills in healthy relationships and safe environments. Developing trust, identity and confidence were important mechanisms for improved mental health. Most interventions focused on the young person's own social support network. The importance of supporting young persons' social networks in order to help them develop skills they require in adulthood has been highlighted as a priority matter in global youth policy [75]. The importance of developing adolescents' social skills and enabling them to improve interpersonal relationships has been identified a central ingredient towards improving their mental health [30,76].

## Strengths and limitations

To our knowledge, this is the first review of social support interventions specifically looking at children's mental health. We applied realist review principles thoroughly and consistently

throughout the research with the aim of generating findings that can guide theoretical thinking around developing programme theories, logic models, and evaluation designs. As with many psychosocial phenomena, there other concepts closely related to social support (such as social connectedness, social capital, loneliness). Investigating one concept but not others will naturally have limitations. For example, it means that we excluded studies in which interventions mobilised or altered social relationships and improved social skills, but did not specifically investigate this from a social support perspective [77]. As typical for realist reviews, the application of inclusion and exclusion criteria was complex. It was difficult to decide whether studies sufficiently conceptualised or measured social support and children's mental health to justify their inclusion. Whilst we sought to address this challenge by adding an additional screening step, we cannot rule out a certain lack of consistency.

## Implications for policy, practice and research

Loneliness and social isolation attract major interest as contributors to poor mental health [78], with young people experiencing loneliness with greatest frequency or intensity of all age groups [79]. Increasing perceived social support, which is considered to be equivalent to reducing loneliness [80,81], might help prevent or reduce mental health problems in young people [82,83]. Few children or young people approach health professionals for help with their mental health problems [84,85] and are instead much more likely to seek help from existing networks of formal or informal supports, such as from teachers and friends [86]. Therefore, interventions seeking to mobilise such networks might have an important role in promoting mental health in this population. However, findings from this review also suggest that, in order for interventions to be effective, they might need to be population- and context-specific, and consider the complex nature of social support. Especially for vulnerable populations who might experience discrimination, lack skills and trust to engage in social relationships, approaches might need to involve changing attitudes towards social support, motivations to engage in social support, and skills to do so. Achieving those changes involves time and resources. As highlighted in a recent review of interventions to reduce loneliness among people with mental health problems [82], it is often unclear whose responsibility it is to invest their time and resources. Social care and community organisations, community (mental) health services and schools are potentially well-placed to actively foster development of informal and formal networks [74,87,88]. However, it also requires policies, strategies and investments that support this kind of systems change. A requirement for a wider roll-out of most interventions includes the knowledge about who should be targeted. Findings from Cacioppo et al (2009) [89] suggest that targeting individuals at the periphery of social networks might have positive knock-on effects for whole communities. Future research and practice developments might be needed to explore how best to identify such children or families at risk of social isolation.

Noticeably, the majority of programme theories in studies identified by our review mobilised parents' social support and focused on improving children's behaviour problems. Less consideration was given to the impact of interventions that mobilise social support to improve child emotional problems, as well as those that mobilise social support networks from the perspective of the child. Additionally, our review only identified one study that included online support. Digital technologies might potentially play important roles in providing social support [90,91]. However, their programme theories are often not detailed in studies highlighting the need for more development work [92].

## Supporting information

**S1 Checklist.**
(DOC)

**S1 Table. Assessment of relevance for included studies.**
(DOCX)

**S2 Table. Assessment of quality of included studies.**
(DOCX)

**S3 Table. Characteristics of included studies concerned with infants (0 to 2 years).**
(DOCX)

**S4 Table. Characteristics of included studies concerned with children (3 to 9 years).**
(DOCX)

**S5 Table. Characteristics of included studies concerned with adolescents (10 to 18 years).**
(DOCX)

**S1 Box. Example of search strategy in PubMed.**
(DOCX)

**S1 File.**
(DOCX)

## Author Contributions

**Conceptualization:** Annette Bauer, Madeleine Stevens, Jean Paul.

**Data curation:** Annette Bauer, Madeleine Stevens, Daniel Purtscheller, Jean Paul.

**Formal analysis:** Annette Bauer, Madeleine Stevens, Daniel Purtscheller, Jean Paul.

**Funding acquisition:** Annette Bauer, Jean Paul.

**Investigation:** Annette Bauer, Madeleine Stevens, Daniel Purtscheller, Peter Fonagy, Jean Paul.

**Methodology:** Annette Bauer, Madeleine Stevens, Jean Paul.

**Project administration:** Daniel Purtscheller, Jean Paul.

**Resources:** Martin Knapp, Sara Evans-Lacko, Jean Paul.

**Software:** Annette Bauer.

**Supervision:** Martin Knapp, Sara Evans-Lacko.

**Validation:** Annette Bauer, Peter Fonagy.

**Writing – original draft:** Annette Bauer.

**Writing – review & editing:** Annette Bauer, Madeleine Stevens, Martin Knapp, Peter Fonagy, Sara Evans-Lacko, Jean Paul.

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
