## [Decision Letter · Decision Letter 0]

12 Apr 2021

PONE-D-20-37767

Mobilising social support to improve mental health for children and adolescents: a systematic review using principles of realist synthesis

PLOS ONE

Dear Dr. Bauer,

Thank you for submitting your manuscript to PLOS ONE. After careful consideration, we feel that it has merit but does not fully meet PLOS ONE’s publication criteria as it currently stands. Therefore, we invite you to submit a revised version of the manuscript that addresses the points raised during the review process.

We look forward to receiving your revised manuscript.

Kind regards,

Veena Kumari

Academic Editor

PLOS ONE

Journal Requirements:

5. Please include captions for your Supporting Information files at the end of your manuscript, and update any in-text citations to match accordingly. Please see our Supporting Information guidelines for more information: http://journals.plos.org/plosone/s/supporting-information

Reviewers' comments:

Reviewer's Responses to Questions

**Comments to the Author**

1. Is the manuscript technically sound, and do the data support the conclusions?

Reviewer #1: Yes

2. Has the statistical analysis been performed appropriately and rigorously? 

Reviewer #1: Yes

3. Have the authors made all data underlying the findings in their manuscript fully available?

Reviewer #1: Yes

4. Is the manuscript presented in an intelligible fashion and written in standard English?

Reviewer #1: Yes

5. Review Comments to the Author

Reviewer #1: This is an interesting review and clearly a lot of work has clearly gone into this SR review. The tables (especially those in the supplement material) are very helpful and add to the current literature.

A few minor suggestions below.

Methods:

Line 177: Study rigor was assessed in relation to study design, samples size, data collection and outcomes, etc. but no detail on what gets you a LOW v HIGH. Please provide a bit more detail on the rating system

Lines 185: I don’t quite understand what this means? Do you mean if the inclusion was 5-12 y/o the study would land in the 3 to 9 category? I think if you remove the “mean” ages from this and just include age range in years given you don’t really calculate a mean.

Grammar: move the common from 2, to after years, and add years after each category)

Age categories included mean ages of infants 0 to 2, years children aged 3 to 9

185 and adolescents aged 10 to 18. For studies, where the age range fell between two

186 categories, they were moved to the category in which the majority of children would fall

187 under.

Lines 189: I don’t quite understand this sentence. What does context-mechanisms outcomes configurations for components of interventions mean? Could you also provide a hypothesized mechanism of change figure or something to make this a bit more clear regarding the relationship you were hoping to evaluate and mechanisms or pathways you were hoping to be explored in the interventions you identified?

By identifying data patterns, a realist synthesis seeks to derive context-mechanisms188

outcomes configurations for components of interventions, which explain how the relationship

189 between resource inputs, human reaction processes, and contextual factors lead to

190 particular outcomes.

Line 197 should be of not if

provide details of the studies including the details of how assessments if study relevance

198 and quality were derived.

Line 281 remove the e.g. before the reference?

review also suggest that in order for interventions to be effective, they might need to

504 population and context specific,

ADD the word BE before population

6. PLOS authors have the option to publish the peer review history of their article (what does this mean?). If published, this will include your full peer review and any attached files.

Reviewer #1: No

---

## [Author Response · Author response to Decision Letter 0]

30 Apr 2021

We included a detailed response to reviewers and editors in a Table which can be found in the response to reviewers document.

---

## [Editor Report · Decision Letter 1]

3 May 2021

Mobilising social support to improve mental health for children and adolescents: a systematic review using principles of realist synthesis

PONE-D-20-37767R1

Dear Dr. Bauer,

We’re pleased to inform you that your manuscript has been judged scientifically suitable for publication and will be formally accepted for publication once it meets all outstanding technical requirements.

Kind regards,

Veena Kumari

Academic Editor

PLOS ONE

Additional Editor Comments (optional):

Thank you very much for revising the paper to carefully incorporate reviewer's comments and feedback.

Best wishes

Veena Kumari
---

## [Editor Report · Acceptance letter]

5 May 2021

PONE-D-20-37767R1 

Mobilising social support to improve mental health for children and adolescents: A systematic review using principles of realist synthesis 

Dear Dr. Bauer:

I'm pleased to inform you that your manuscript has been deemed suitable for publication in PLOS ONE. Congratulations! Your manuscript is now with our production department. 

Kind regards, 

on behalf of

Dr. Veena Kumari 

Academic Editor

PLOS ONE